# Abnormal LAMP1 glycosylation may play a role in Niemann-Pick disease, type C pathology

**Niamh X. Cawley** [1‡], **Caitlin Sojka** [1‡], **Antony Cougnoux**[1], **Anna T. Lyons**[1], **Elena-Raluca Nicoli** [2], **Christopher A. Wassif**[1], **Forbes D. Porter**[1] *

**1** Section on Molecular Dysmorphology, Division of Translational Medicine, Department of Health and Human Services, *Eunice Kennedy Shriver* National Institute of Child Health and Human Development, National Institutes of Health, Bethesda, MD, United States of America, **2** Department of Health and Human Services, Pediatric Undiagnosed Diseases Program, National Human Genome Research Institute, National Institutes of Health, Bethesda, MD, United States of America

‡ These authors are Joint first authors on this work.
* fdporter@mail.nih.gov

**Data Availability Statement:** All relevant data are within the manuscript and its Supporting Information files.

## Abstract

A hallmark of Niemann-Pick disease, type C (NPC) is the progressive degeneration of Purkinje neurons in the cerebellum caused by the accumulation of free cholesterol and glycosphingolipids in the lysosome. Recent studies suggest that the state of glycosylation of lysosomal membrane proteins may play a role in disease progression. Our study has identified the presence of a highly glycosylated form of Lysosome Associated Membrane Protein 1 (LAMP1) that correlated spatiotemporally with Purkinje neuron loss. This form of LAMP1 was predominantly localized to activated microglia; showing a ~5-fold increase in surface labeling by FACS analysis. This suggests a potential role for LAMP1 in the neuro-inflammatory process in these mice during disease progression. Analysis of other mouse models of neurodegeneration that exhibit neuro-inflammation showed little or no presence of this glycosylated form of LAMP1, suggesting this observation for LAMP1 is specific to NPC disease. Furthermore, early treatment of $Npc1^{-/-}$ mice with 2-hydroxypropyl-β-cyclodextrin (HPβCD), significantly prevented the appearance of the glycosylated LAMP1 in the cerebellum of $Npc1^{-/-}$ mice at 7 weeks, consistent with the prevention of neuro-inflammation in mice treated with this drug. Treatment of $Npc1^{-/-}$ mice with HPβCD at 7 weeks, after disease onset, did not reverse or prevent further appearance of the hyperglycosylated LAMP1, demonstrating that once this aspect of neuro-inflammation began, it continued despite the HPβCD treatment. Analysis of LAMP1 in cerebellar tissue of NPC1 patients showed a small level of hyperglycosylated LAMP1 in the tissue, however, this was not seen in the CSF of patients.

**Funding:** Funding for this work was from the Intramural Research Program of the Eunice Kennedy Shriver National Institute of Child Health and Human Development

**Competing interests:** The authors have declared that no competing interests exist.

## Introduction

Niemann-Pick disease, type C (NPC) is a fatal, neurodegenerative disorder characterized by an accumulation of un-esterified cholesterol and glycosphingolipids in endo-lysosomal compartments of cells [1, 2]. The disease is autosomal recessive, where patients carry mutations in either *NPC1* or *NPC2* that encode for proteins normally involved in the binding and transport of unesterified cholesterol out of the lysosome [3–5]. The cholesterol can then be distributed to other cellular organelles [4, 6, 7]. The disease affects ~1:120,000 births [2] and is classified as a rare lysosomal storage disorder (LSD) [8, 9] that ultimately results in early death of the patient.

Lysosomes are membrane-bound cellular organelles containing hydrolytic enzymes that are responsible for the degradation of proteins and lipids for re-use within the cell. Membrane proteins of the lysosome play a role in this break-down process, in addition to being involved with the trafficking and fusion of lysosomes with other compartments of the endosomal/lysosomal pathway [10–12]. One major group of these proteins are lysosome-associated membrane proteins (LAMPs); type I integral membrane proteins with heavily glycosylated luminal domains and short cytosolic tails [13]. LAMP1 and LAMP2, which are similar in size and structure, together make up about 50% of all lysosomal membrane proteins [11], however, their exact role in the function of the lysosome is complex [14, 15]. Mice deficient in LAMP1 (*Lamp1$^{-/-}$*) appear to be normal, as they do not exhibit any significant global differences compared to control mice (*Lamp1$^{+/+}$*) [16]. However, at the tissue level, mild astrogliosis in the neocortex is observed, and a compensatory increase in LAMP2 protein is seen primarily in the kidney. In contrast, LAMP2 knock out (KO) mice (*Lamp2$^{-/y}$*) exhibit a completely different phenotype, suggesting the two proteins may have distinct functions. *Lamp2$^{-/y}$* mice recapitulate Danon disease, an X-linked lysosomal storage disorder characterized by (cardio)myopathy and intellectual dysfunction [17, 18]. Indeed, hippocampal dysfunction due to inflammation, and accumulation of autophagic vacuoles and lipid storage in neurons, reminiscent of NPC neuropathology [2], may provide insight into the neuropathological phenotype observed in these KO mice. Parenthetically, double mutant Lamp1$^{-/-}$/Lamp2$^{-/y}$ mice are embryonically lethal. Embryonic fibroblasts from these mice accumulate cholesterol which can be rescued by overexpression of *Lamp2*, but not *Lamp1* [19], further suggesting a divergence in function of these two similar proteins.

Both LAMP1 and LAMP2 are heavily glycosylated on their luminal domains; predominantly composed of N-linked glycans, N-acetylglucosamine (GlcNAc) linked to asparagine residues. In addition, O-linked glycans, N-acetylgalactosamine (GalNAc), are linked to serine or threonine residues [20, 21]. The glycan structures form a lining on the inner leaflet of the lysosome, known as the glycocalyx, an approximately 8 nm barrier [15], that can protect the LAMPs and other membrane components of the lysosome from degradation by hydrolytic enzymes [14]. Recently, several studies demonstrate that LAMP1 glycosylation is altered in NPC1 and may contribute to the accumulation of cholesterol and glycosphingolipids [8, 9]. Treatment of NPC1 fibroblasts with an inhibitor of O-linked glycosylation, reduced cholesterol storage and increased cholesterol efflux from the cells [8], suggesting that the reduced glycosylation of LAMP1 could be beneficial in reducing the NPC1 disease phenotype. In addition, a recent report identified a key enzyme in the O-linked glycosylation pathway, β3GALT5, that is differentially expressed across the lobules of the cerebellum [22], suggesting a possible difference in the glycosylation complexity of LAMP1 in the lobules that are resistant to neurodegeneration. Considering these observations, we undertook to investigate the physical characteristics of LAMP1 and LAMP2 in the cerebellum of *Npc1$^{-/-}$* mice during disease progression. While we found increased expression levels for both proteins, a highly glycosylated

form of LAMP1 was spatially and temporally correlated with disease progression. Our data suggest that abnormal glycosylation of LAMP1 may play a role in NPC pathology.

## Materials and methods

### Animal maintenance and tissue collection

All animal work conformed to the National Institutes of Health guidelines and was approved by the *Eunice Kennedy Shriver* National Institute of Child Health and Human Development Institutional Animal Care and Use Committee (Protocol #18–002). BALB/c-*Npc1*$^{+/-}$, BALB/c-*Npc2*$^{+/-}$, and C57BL/6J- *SOD1*$^{G93A}$ (Cu/Zn superoxide dismutase 1) mice were obtained from The Jackson Laboratory (Bar Harbor, ME). Mucolipidosis type IV mice, C57BL6-*Mcoln1*$^{+/-}$, were a gift from Dr. James Pickel (NIMH, NIH) and Fabry mice, C57BL6-*Gla*$^{-/-}$, were a gift from Dr. Ashok Kulkarni (NIDCR, NIH) [23, 24]. Adult experimental autoimmune encephalitis (EAE) mice, sacrificed at 12 weeks of age, were a gift from Dr. Keiko Ozato (NICHD, NIH) [25]. Cerebellar tissue from ASM mice was generously provided by Dr. Steven Walkley (Albert Einstein College of Medicine, NY). Cerebellar tissue from Sandhoff (*Hexb*$^{tm1Rlp}$) [26] and GM1 (*Glb1*) mice, was generously provided by Dr. Cindy Tifft (NHGRI, NIH). Pups were weaned 3 weeks after birth and subsequently had free access to water and normal mouse chow. PCR genotyping was performed using tail or ear DNA. *Npc1*$^{-/-}$ mice were sacrificed at 3, 5, 6, 7, and 9 weeks of age, while all other mouse models were sacrificed at the respective peak of disease severity. For drug treatment, mice were injected (SQ) with Kleptose HPB (2-hydroxy-propyl-β-cyclodextrin (HPβCD); VTS-270; 4000 mg/kg) (Roquette, Vacquemont, France) or the vehicle control (PBS), every 3 days starting at either 21 days (pre-symptomatic) or 49 days (symptomatic), until sacrifice and analysis as indicated.

### Immunofluorescence and cerebellum micro-dissection

Control and mutant mice (7–9 weeks of age) were euthanized by $CO_2$ asphyxiation and transcardially perfused with room temperature (RT) phosphate buffered saline (PBS). Whole brain was removed and either collected in 4% paraformaldehyde (PFA) in PBS, pH 7.4 for immunofluorescence (see below) or the cerebellum was removed, and flash frozen on dry ice for subsequent micro-dissection. For micro-dissection, the hemispheres were removed using a cold razor blade prior to a midline sagittal cut to split the vermis in half. Each half vermis was then placed on its side in 100 µl of ice-cold PBS and micro-dissected under a dissection microscope. Lobules I to V were collected as the anterior region, lobules VI to VIII as the middle region and lobules IX and X as the posterior region.

For immunofluorescence, the brains were post-fixed in 4% PFA solution for 24 h, then cryoprotected in 30% sucrose. Cerebellar tissues were cryostat-sectioned parasagittally (20 µm) and floating sections collected in PBS containing 0.25% Triton X-100. Sections were blocked for 30 min in 10% normal goat serum (Sigma-Aldrich, St-Louis, MO, USA) and incubated overnight at 4˚C with primary antibody. Primary antibodies included: rabbit anti-calbindin (1:400, Cell Signaling Technology, Danvers, MA, USA), anti-GFAP (1:500, Novus Biologicals, LLC, Littleton, CO, USA), anti-Iba1 (1:500, Wako Pure Chemicals Industries Osaka, Japan), anti-Olig-2 (1:200, Millipore Sigma, Billerica, MA, USA), anti-LAMP1 (1:200, DSHB, Iowa City, IA, USA) and anti-LAMP1-Alexa-488 (1:100, Biolegend, San Diego, CA). Secondary antibodies include Alexa Fluor 594 goat anti-rat IgG, Alexa Fluor 488 goat anti-rabbit IgG, Alexa Fluor 488 goat anti-mouse IgG, and Alexa Fluor 488 goat anti-chicken IgY (1:1000, Thermo Fisher Scientific, Waltham, MA, USA). Hoechst 33342, Trihydrochloride, Trihydrate I (1:1000, Invitrogen, Carlsbad, CA, USA) was used to stain the nuclei. Images were taken on

either a Zeiss Axio Observer Z1 microscope (Zeiss, Germany) or a Zeiss 780 confocal microscope (NICHD Microscopy Core Facility).

## Protein extraction and Western blots

Total protein was extracted from cells and tissues using ice cold T-PER Tissue Protein Extraction Reagent (Thermo Scientific, USA) supplemented with 1% Triton X-100, and 1X Mammalian ProteaseArrest inhibitor cocktail (G-Biosciences, St. Louis, MO, USA). Protein quantification was performed using the DC protein assay (Bio-Rad Laboratories, Hercules, CA, USA). Protein lysates (typically 10–20µg) were analysed by Western blot on 4–12% gradient or 12% NuPAGE gels (Invitrogen, Carlsbad, CA, USA). Protein transfer was performed using the iBLOT 2 dry transfer apparatus (Life Technologies Carlsbad, CA, USA) according to the manufacturer's protocol. Following transfer, nitrocellulose membranes were incubated in a 1X PBS/0.1% Tween 20 blocking buffer with 5% non-fat milk for 1 h at RT, followed by incubation in the primary antibody overnight at 4˚C. After the initial incubation, the membrane was rinsed with 1X PBS/0.1% Tween 20 and incubated with the appropriate secondary antibody for 1 h at RT. Membrane development was carried out using either the Bio-Rad chemiluminescence detection kit or the LI-COR Biosciences Odyssey CLx Imaging System. Primary antibodies used include: anti-LAMP1 (1:2,000, Cell Signalling Technology, Danvers, MA, USA), anti-β-actin (1:2,000, Sigma Aldrich, St. Louis, MO, USA), anti-Glutamine Synthetase (1:1,000, Abcam, Cambridge, MA, USA), anti-β-III Tubulin (1:1,000, Abcam, Cambridge, MA, USA), anti-LAMP2 (1:1,000, Clone ABL-93, Developmental Studies Hybridoma Bank, University of Iowa), anti-LIMP2 (1:1000, Thermo Fisher Scientific, Waltham, MA, USA), anti-GFAP (1:1000, Novus Biologicals, Littleton, CO, USA). Secondary antibodies used include: anti-mouse IgG-HRP or anti-rabbit IgG-HRP (1:10,000, Sigma Aldrich, St. Louis, MO, USA), goat-anti-chicken IgY(IgG) (H+L)-HRP conjugate (1:10,000, Advansta, Menlo Park, CA, USA), IRDye 800CW goat anti-rabbit IgG (H + L) (1:15,000, LI-COR, Lincoln, NE, USA). For quantitation, band intensities (integrated adjusted signal) were normalized to β-actin using Image Studio Lite Ver 5.2 software (LI-COR Biosciences, Lincoln, NE, USA).

## Deglycosylation of LAMP1

Whole cerebellum protein lysate was extracted, as described above, from age-matched 7-week wild type ($Npc1^{+/+}$) and mutant ($Npc1^{-/-}$) mice. N-glycanase, an asparagine amidase that cleaves the Asn-GlcNac bond of N-glycans, and Endoglycosidase H (Endo H), a glycosidase that cleaves within the chitobiose core of complex mannose and some hybrid oligosaccharides (New England Biolabs, Inc., Ipswich, MA, USA), were used to characterize the glycosylation complexity of LAMP1. Cerebellum protein lysates were treated with and without N-glycanase or Endo H in duplicates, according to the kit protocol (New England Biolabs). The treated lysates were analyzed by Western blot for LAMP1.

## Immunoprecipitation

Whole cerebellum was used from age-matched 7-week control ($Npc1^{+/+}$) and mutant ($Npc1^{-/-}$) mice. For primary microglia isolation, tissue was dissociated and prepared using papain, following the manufacturer's protocol (Worthington Biochemical Corporation, Lakewood, NJ, USA). The anti-CDllb (microglia) MicroBead kit for human and mouse, was used to isolate primary microglia from whole cerebella following the manufacturer's protocol (Miltenyi Biotec Inc., Auburn, CA, USA). Samples were analyzed by Western blot for LAMP1, CD11b and glutamine synthase.

## Fluorescence-activated cell sorting

Mice were euthanized in a $CO_2$ chamber and then transcardially perfused with 25 ml 1X PBS. The cerebellum was recovered from the brain and suspended in ice cold FACS buffer (PBS containing 0.5% BSA (Sigma)). Tissue samples were gently homogenized on a sterile culture dish and collected in 7 ml FACS buffer. A discontinuous Percoll gradient 30–70% (GE Health-care) was prepared and samples were centrifuged 2,450 rpm for 30 minutes at 4˚C. Cells at the 70% interface were collected, washed twice in FACS Buffer and stained for live\dead cells using the Violet live\dead dye (1:500, Life Technology), CX3CR1 (1:25, R&D), CD11b (1:100, Biolegend), CD45 (1:100, Biolegend). Live, triple positive cells were isolated using FACS Aria 3 (BD Biosciences), performed by the NHLBI, NIH core facility. For select experiments, cells were also stained for LAMP1 (1/100, Biolegend) and cells positive for all four markers were isolated.

## Lipopolysaccharide treatment

Microglia were isolated from $Npc1^{+/+}$ mice as described above. Cells were immediately plated in 12-well culture dish (BD, San Diego, CA, USA) and left overnight in 37ºC, 5% $CO_2$ in a humidified incubator. The following day, the DMEM growth medium (Life Technology, Carlsbad, CA, USA), supplemented with 10% FBS (Omega Scientific, Offenburg, Germany), was changed. Six hours post-medium change, the cells were rinsed twice with 1X PBS, and fresh medium containing 1μg/ml Escherichia coli lipopolysaccharide O55:B5 (LPS, Invitrogen, USA) was added to the cells. After a 60-min incubation in the LPS-medium, cells were rinsed twice with 1X PBS and harvested for analysis by Western blot.

## Human samples

Human post-mortem cerebellar samples from NPC1 patients were obtained from University of Maryland Brain and Tissue Bank (https://www.medschool.umaryland.edu/btbank/). All NPC1 CSF samples were collected by lumbar puncture in the L4/L5 interspace of patients who were enrolled in an *Eunice Kennedy Shriver* National Institute of Child Health and Human Development Institutional Review Board-approved longitudinal Natural History observational trial at the National Institutes of Health between August 2006 and January 2011 (06-CH-0186, NCT00344331). Cerebrospinal fluid from five females and five males of varying age and severity was used in this study. Single donor normal human CSF from five females and five males was purchased from Innovative Research, Novi, MI. All samples were stored at -80˚C prior to being assayed.

## Concentration of CSF for Western blot analysis

Cerebrospinal fluid (CSF, 270 μl) proteins were precipitated with 10% trichloroacetic acid (TCA) for ~18 h at 4˚C, centrifuged for 10 min at 10,000 x g, and the pellet reconstituted in 30 μl 1X SDS-sample buffer. An aliquot of this was analyzed by Coomassie staining after separation on a 4–12% NuPage gel (Invitrogen), and total protein per lane was quantified by the Odyssey infra-red (IR) scanner in the 700 nm channel. In parallel, an aliquot was analyzed by standard Western blotting procedures for LAMP1 detection, using rabbit-anti-LAMP1 (1: 2,000, Cell Signaling) and the Odyssey IR dye labeled secondary anti-rabbit-800 antibodies (1: 10,000, LI-COR). Integrated signals for both the Coomassie stained proteins (700 nm channel) and the immuno-detected LAMP1 (800 nm channel) were used to calculate the relative levels of LAMP1/protein per CSF sample.

### Statistical analyses

Statistical analyses were done using R/R Studio software (Boston, MA, USA). P-values < 0.05 were deemed statistically significant.

## Results

### NPC1 mouse central nervous system tissues have two distinct forms of LAMP1

Initial Western blot analysis of tissues from control mice ($Npc1^{+/+}$) showed a single ~75 kDa-sized LAMP1 protein in CNS tissues compared to the expected size of 90–120 kDa sized-LAMP1 seen in the peripheral tissues (Fig 1A). In comparison to $Npc1^{+/+}$ mice, CNS tissues of $Npc1^{-/-}$ mice showed an overall increase in the ~75 kDa LAMP1 expression as well as the appearance of an additional larger, more diffuse LAMP1 band at ~90–120 kDa in size (Fig 1B, top panel). Analysis of two other lysosomal membrane proteins, LAMP2 and LIMP2, also showed a general increase in expression levels in $Npc1^{-/-}$ CNS tissues, however, there was no apparent higher molecular mass band observed for these proteins when compared to $Npc1^{+/+}$ tissues (Fig 1B, middle and lower panels). The size estimate of ~75 kDa was based on its position on Western blots compared to standards and an endogenous marker HSP70 (S1 Fig).

### Larger LAMP1 appears in the same spatiotemporal pattern as Purkinje neuron degeneration in NPC1 mouse cerebella

In $Npc1^{-/-}$ cerebellar extracts, Western blot analysis showed that the larger form of LAMP1 appears first between 5 and 7 weeks of age, whereas in $Npc1^{+/+}$ cerebellar extracts only the ~75

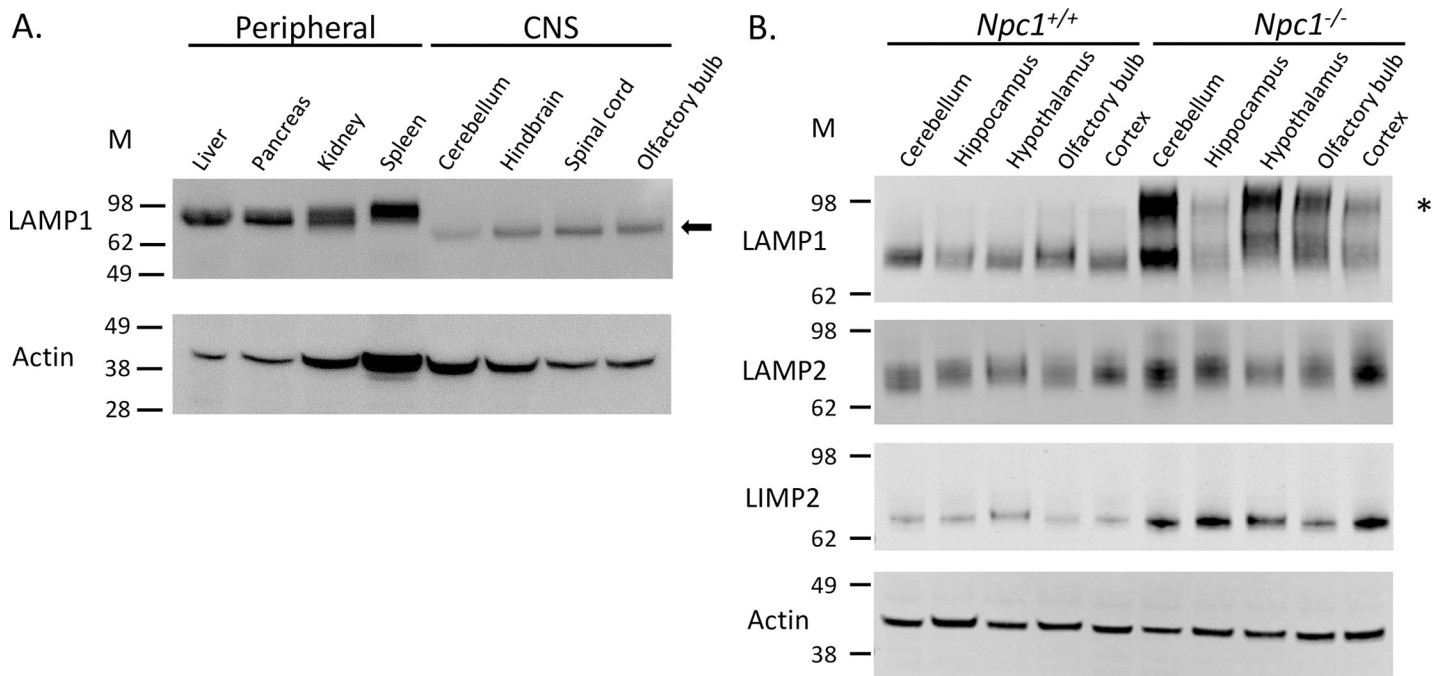

**Fig 1. Lysosomal membrane proteins in NPC1 tissues. A.** Western blot of 7-week $Npc1^{+/+}$ peripheral tissues (liver, pancreas, kidney, spleen) and CNS tissues (cerebellum, hindbrain, spinal cord, olfactory bulb) for LAMP1 and actin (loading control). **B.** Western blot of CNS tissues from 9-week $Npc1^{+/+}$ and $Npc1^{-/-}$ mice for LAMP1, LAMP2, and LIMP2. β-III tubulin was used as a loading control. The asterisk (*) marks the larger size of LAMP1 that is referenced as hyperglycosylated LAMP1 in the text. Arrow indicates LAMP1 size of ~75 kDa in CNS tissue (see S1 Fig for size estimate).

kDa form is present (Fig 2A). The appearance of this larger protein correlates with the onset of phenotypic disease progression of NPC1. Further analysis showed that the larger form relative to the smaller form of LAMP1 in $Npc1^{-/-}$ appears first in the anterior region of the cerebellum at 6 weeks of age compared to the posterior region (Fig 2B). This also correlates with the disease-related Purkinje neuron death pattern, which occurs in the cerebellum of NPC1 in an anterior to posterior manner at these ages [1].

A two-way ANOVA compared the ratios of the upper to the lower LAMP1 band intensities across age (5, 6, 7 weeks) and cerebellar lobule region (anterior, middle, posterior). The ratio of the upper to the lower LAMP1 signal is significantly increased across ages (p<0.001). In addition, the intensity of the larger LAMP1 band tended to be differentially expressed in an anterior to posterior manner, most noticeably at 6 weeks of age, although this did not reach statistical significance (p = 0.055; Fig 2C).

## The larger LAMP1 protein in $Npc1^{-/-}$ cerebellum contains complex N-linked glycosylation

Treatment of total cerebellar extracts from $Npc1^{+/+}$ and $Npc1^{-/-}$ mice with N-glycanase followed by Western blot analysis showed that all forms of LAMP1 present in this tissue were

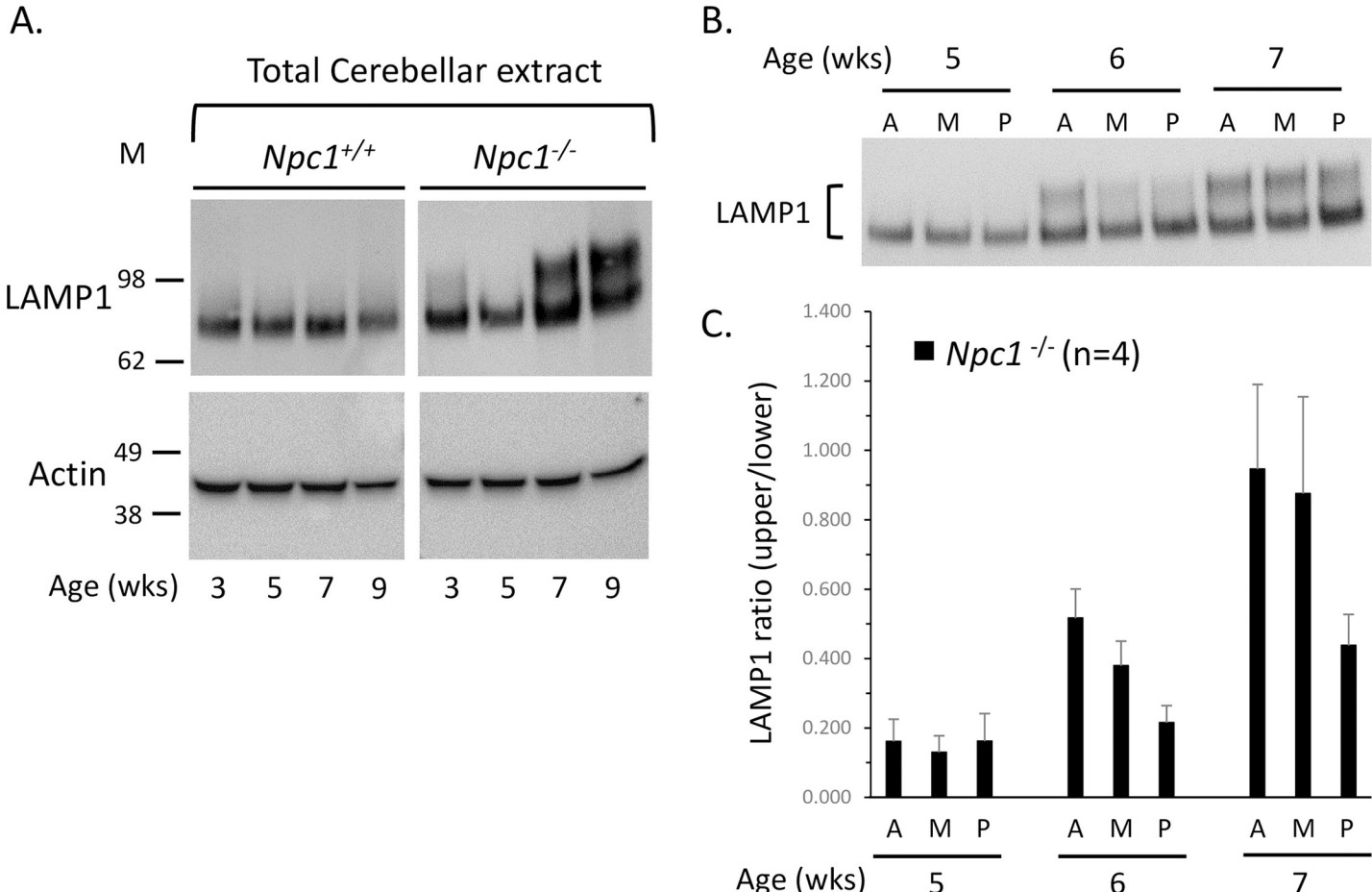

**Fig 2. Spatiotemporal expression pattern of LAMP1. A.** Western blot of total cerebellar extract from $Npc1^{+/+}$ and $Npc1^{-/-}$ mice at 3, 5, 7, and 9 weeks of age, for LAMP1 and actin (loading control). **B.** Western blot of anterior (A), middle (M), and posterior (P) regions of cerebellum from $Npc1^{-/-}$ mice at 5, 6, and 7 weeks of age, for LAMP1. **C.** The Western blot in (B) was replicated with different mice (n = 4) and the LAMP1 ratio (intensity of the higher molecular weight band to the lower molecular weight band) was calculated. The means of LAMP1 ratios were graphed. Two-way ANOVA indicates a significant difference (P<0.001) in LAMP1 ratio across age, but only a trend across cerebellar region (P = 0.055). There is no significant interaction between age and cerebellar region (P>0.05).

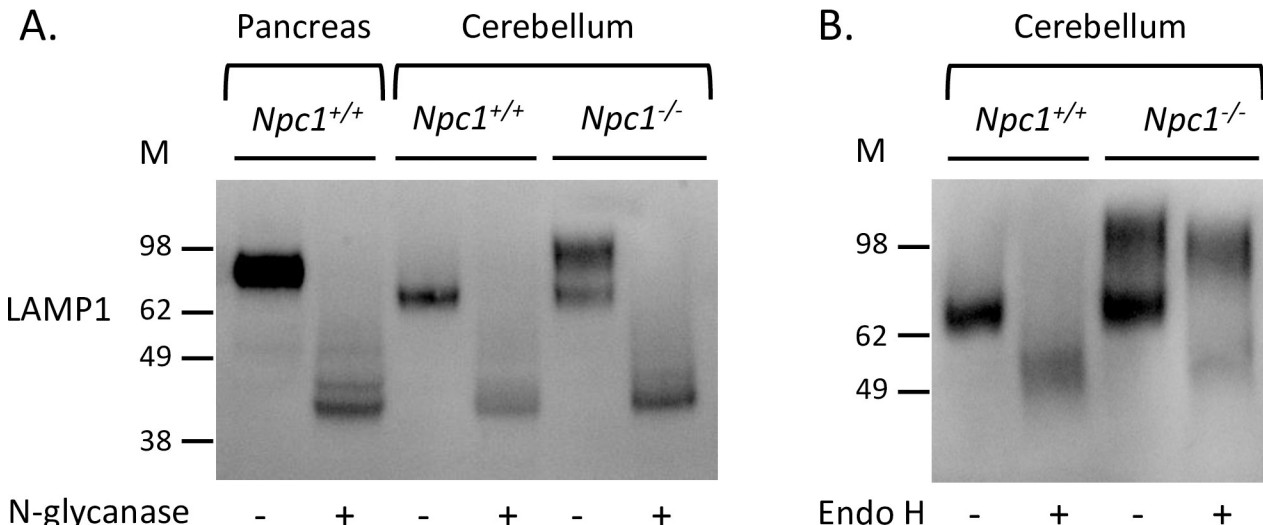

**Fig 3. Higher molecular weight is due to LAMP1 hyperglycosylation. A.** LAMP1 Western blot of 9-week *Npc1*<sup>+/+</sup> pancreas and cerebellum and *Npc1*<sup>−/−</sup> cerebellum extract treated with (+) and without (-) N-glycanase. **B.** LAMP1 Western blot of 9-week *Npc1*<sup>+/+</sup> and *Npc1*<sup>−/−</sup> cerebellum extract treated with (+) and without (-) Endo H.

susceptible to N-linked deglycosylation (Fig 3A). The resulting deglycosylated LAMP1 is ~40 kDa in size; a size consistent with the size of the protein backbone of the LAMP1 [27]. Furthermore, this suggests that LAMP1 is differentially N-glycosylated in the cerebellum during disease progression of NPC1. In contrast to N-glycanase treatment, extracts treated with endoglycosidase H (Endo H) showed that only the ~75 kDa LAMP1 and a small amount of the larger LAMP1 were susceptible to this form of deglycosylation (Fig 3B). The larger LAMP1 form was predominantly resistant to Endo H, demonstrating that the larger form of LAMP1 (now referred to as hyperglycosylated LAMP1) contained complex or hybrid N-linked glycosylation. Analysis of LAMP1 from other areas of the brain of *Npc1*<sup>−/−</sup> mice gave similar results (S2A Fig). Of note was the observation that LAMP1 from peripheral tissue was also Endo H resistant (S2B Fig), suggestive of a peripheral LAMP1 glycosylation phenotype in the CNS of affected NPC1 mice. The overall weaker signal for LAMP1 in the treated samples may reflect a loss of protein due to low level protease degradation or a reduced antibody affinity.

## Immunofluorescence (IF) and immunoprecipitation (IP) show that hyperglycosylated LAMP1 is associated with microglia

To investigate the cellular localization of the hyperglycosylated LAMP1, IF was performed on *Npc1*<sup>+/+</sup> and *Npc1*<sup>−/-</sup> cerebella. Cerebellar sections were co-stained for LAMP1 and several different cell markers including: Iba1 (microglia), Calbindin 1 (Purkinje neurons), Olig2 (oligodendrocytes), GFAP (reactive astrocytes), and S100β (Bergmann glia). In *Npc1*<sup>+/+</sup> cerebellum, LAMP1 is present in a distinct pattern predominantly aligned with the Purkinje neuron layer (S3 Fig). Higher magnification shows apparent staining of LAMP1 in Purkinje neuron soma as well as other cells surrounding the Purkinje neurons (S4A Fig). In contrast to LAMP1 staining in *Npc1*<sup>+/+</sup> cerebellum, the distribution of LAMP1 in *Npc1*<sup>−/-</sup> is more disorderly. Its distinct pattern is disrupted as the Purkinje neurons degenerate (S4B Fig). In addition, a more widespread and stronger staining pattern of LAMP1, that is not restricted to the Purkinje neuron layer, is seen throughout all the lobules of the cerebellum. Confocal microscopy of the cerebellum confirmed that LAMP1 immunostaining was present in the calbindin-positive soma of

the Purkinje neurons from *Npc1*$^{+/+}$ cerebellum (Fig 4A). In addition, LAMP1 also co-stained with S100β positive cells of *Npc1*$^{+/+}$ mice suggesting expression of LAMP1 in Bergmann glia (Fig 4C). In *Npc1*$^{-/-}$ cerebellum, LAMP1 immunostaining increased throughout the lobule and was present in residual Purkinje neurons (Fig 4B). There was an overall increase in S100β staining in the *Npc1*$^{-/-}$ cerebellum consistent with the increase in astrogliosis in the disease state (Fig 4D).

In *Npc1*$^{-/-}$ cerebellum, LAMP1 strongly co-stains with Iba1, a surface marker for microglia (Fig 5A). There was no apparent co-staining of LAMP1 with Olig2 (Fig 5B), a marker for

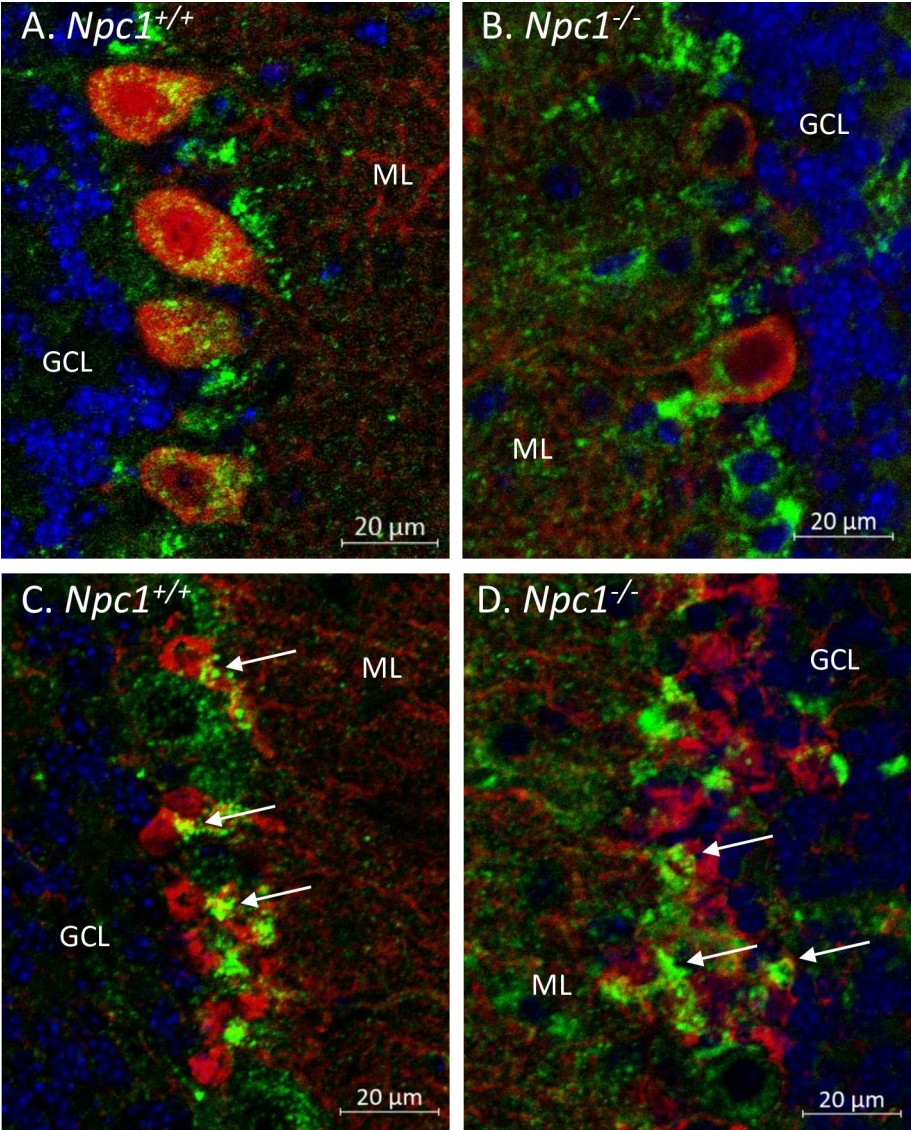

**Fig 4. Confocal immunofluorescence microscopy of 7-week-old *Npc1*$^{+/+}$ and *Npc1*$^{-/-}$ lobule I cerebellum.** (**A**) LAMP1 staining (green) is present in the soma of Purkinje neurons (red) and other cells in the Purkinje cell layer of *Npc1*$^{+/+}$ cerebellum. (**B**) There is a dramatic loss of Purkinje neurons in *Npc1*$^{-/-}$ cerebellum at this age and a general increase in LAMP1 staining. (**C**) Other LAMP1 positive cells were identified as Bergmann glia by their localization and co-staining with S100β (arrows). (**D**) Increased S100β staining is observed in *Npc1*$^{-/-}$ consistent with an increase in astrogliosis. Sections were counterstained with Hoechst 3342 (blue), distinguishing the granule cell layer (GCL) from the molecular layer (ML).

**Fig 5. LAMP1 co-staining in the cerebellum.** Immunohistochemistry of midline sagittal cerebellar sections of 9-week $Npc1^{-/-}$ mice with LAMP1 (red) and different cerebellar cell markers (green), including: Iba1 (**A**), Olig2 (**B**), and GFAP (**C**), counterstained with DAPI (blue). Closed-arrows marks Olig2-positive oligodendrocytes (green). Cerebellar lobule numbers are indicated.

oligodendrocytes and only weak co-staining with GFAP, a marker of reactive astrocytes including Bergmann glia (Fig 5C).

To confirm the presumed localization of LAMP1 in microglia, magnetic bead immunoprecipitation (IP) was used to isolate specific cell types. Intact microglia (CD11b+) from whole cerebellar extracts of 7-week old $Npc1^{+/+}$ and $Npc1^{-/-}$ mice were isolated. Western blot analysis of these samples indicated that all the hyperglycosylated LAMP1 in $Npc1^{-/-}$ cerebellum is present in the immunopurified CD11b+ microglia (Fig 6A). Similar results were observed for the analysis of hyperglycosylated LAMP1 in the cerebella of $Npc2^{+/-}$ and $Npc2^{-/-}$ mice (S5A Fig). As a control, the hyperglycosylated LAMP1 was not present in immunopurified astrocytes when performed on $Npc2^{-/-}$ cerebella (S5B Fig). These results collectively suggest that the hyperglycosylated form of LAMP1 in the diseased state for NPC1 and NPC2 is almost exclusively in activated microglia. Indeed, purified microglia from control mice in culture exhibited only the hyperglycosylated LAMP1 form in these cells (S6 Fig). Additionally, treatment of these cells in culture with Lipopolysaccharide (LPS), to induce microglia activation, did not result in the appearance of additional LAMP1 forms, although the slightly more diffuse nature of the band would suggest a more heterogenous level of glycosylation upon activation (S6 Fig).

Although LAMP1 is a lysosomal protein, it has been reported that LAMP1 may also reside on the plasma membrane of some cells and play a role in cellular migration [27–29]. To test this, fluorescence activated cell sorting (FACS) of cerebellar cells, positive for three microglial surface markers: CX3CR1$^+$, CD11b$^+$, CD45$^{low}$, along with LAMP1, was performed. The results showed a ~5-fold increase in the presence of LAMP1 on the surface of the activated microglia isolated from $Npc1^{-/-}$ mice compared to those from control $Npc1^{+/+}$ mice (Fig 6B).

## Hyperglycosylated LAMP1 is associated with the neuro-inflammatory phenotype of NPC

Activated microglia is a hallmark of neuro-inflammation in the CNS. To determine if the hyperglycosylated LAMP1 observed on activated microglia of NPC1 was specific to the neuro-inflammation associated with NPC, Western blot analysis was performed on affected CNS tissues from several mouse models of neurodegeneration at the ages where neuro-inflammation is present; Niemann-Pick, type C1 and C2 (*Npc1*, *Npc2*), Niemann-Pick, Type A/B (Acid Sphingomyelinase (ASM); *Smpd1*), Sandhoff (*Hexb^{tm1Rlp}*), GM1 (*Glb1*), Mucolipidosis Type IV (ML4; *Mcoln1*), Experimental Autoimmune Encephalitis (EAE), and a model of Amyotrophic Lateral Sclerosis (ALS) (*SOD1^{G93A}*). As a negative control, tissue from a mouse model of Fabry disease (*Gla*), a lysosomal storage disease that does not exhibit neuro-inflammation, was

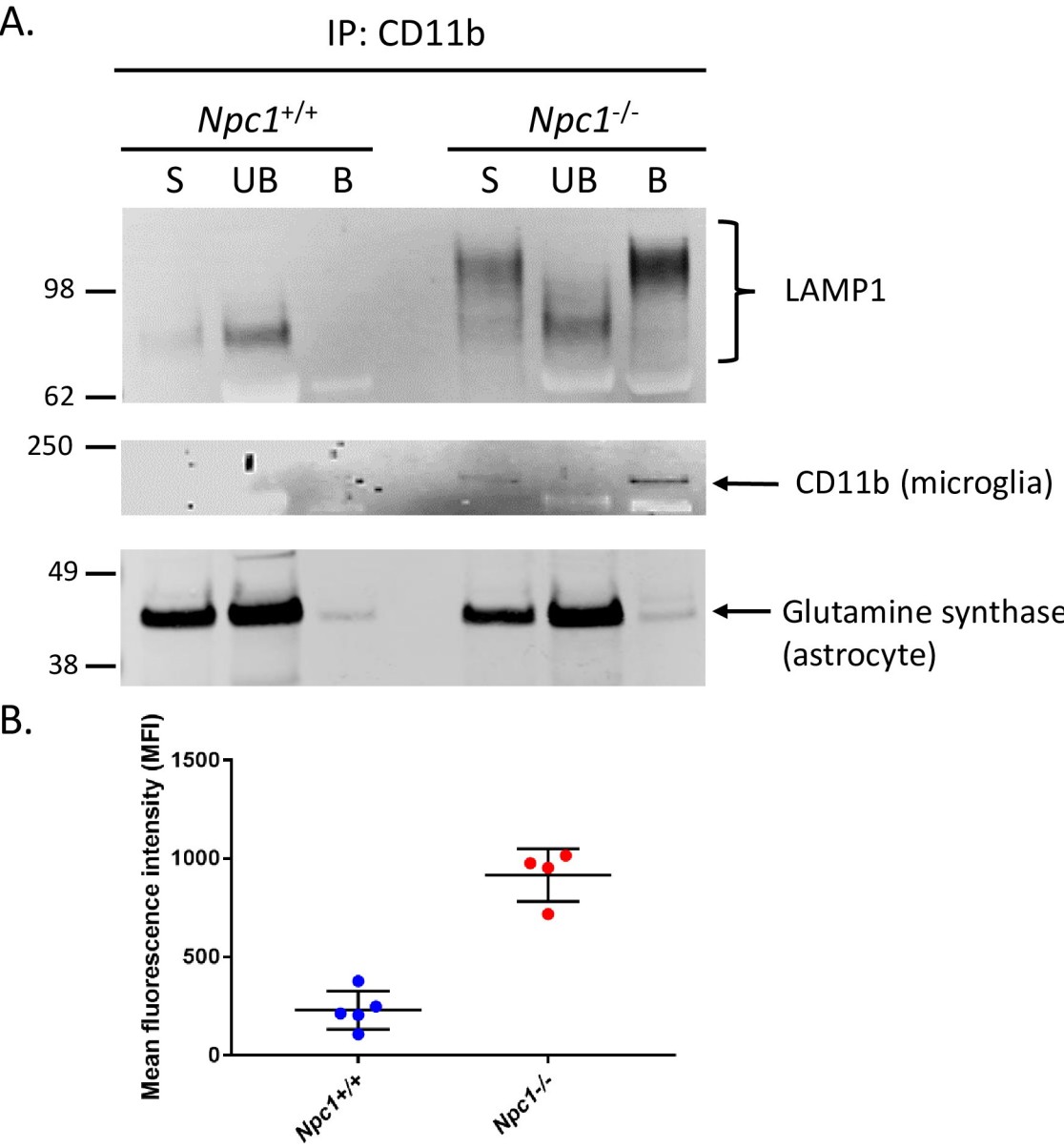

**Fig 6. Hyperglycosylated *Npc1*$^{-/-}$ LAMP1 is present predominantly in microglia.** (**A**) LAMP1 Western blot of cell lysate from 7-week *Npc1*$^{+/+}$ and *Npc1*$^{-/-}$ mouse cerebella that underwent tissue dissociation and subsequent CD11b immunoprecipitation. **A.** Lanes show 10% of the un-bound (UB) and bound (B) fractions of CD11b immunoprecipitation (microglia) reaction. An aliquot representing 3% of the starting material is indicated by S. **B.** FACS analysis of activated cerebellar microglia. Shown is the mean fluorescence intensity (MFI) of LAMP1 for *Npc1*$^{+/+}$ and *Npc1*$^{-/-}$ cerebellar cells that were triple-positive for: CX3CR1, CD11b, CD45.

also analyzed. The results showed that the distinct hyperglycosylated form of LAMP1 is predominantly in samples from affected *Npc1*$^{-/-}$ and *Npc2*$^{-/-}$ mice (cerebellum) (Fig 7A and S5 Fig), and to a lesser extent in the affected Sandhoff and GM1 mice (Fig 7B). A small amount was observed in the ALS mice (Fig 7C) (spinal cord) and no distinct hyperglycosylated LAMP1 was observed in the ASM (Fig 7A), ML4 (hippocampus) or EAE (cerebellum) mouse tissues analyzed. As expected, cerebellum from affected Fabry mice was negative for hyperglycosylated LAMP1 (Fig 7C).

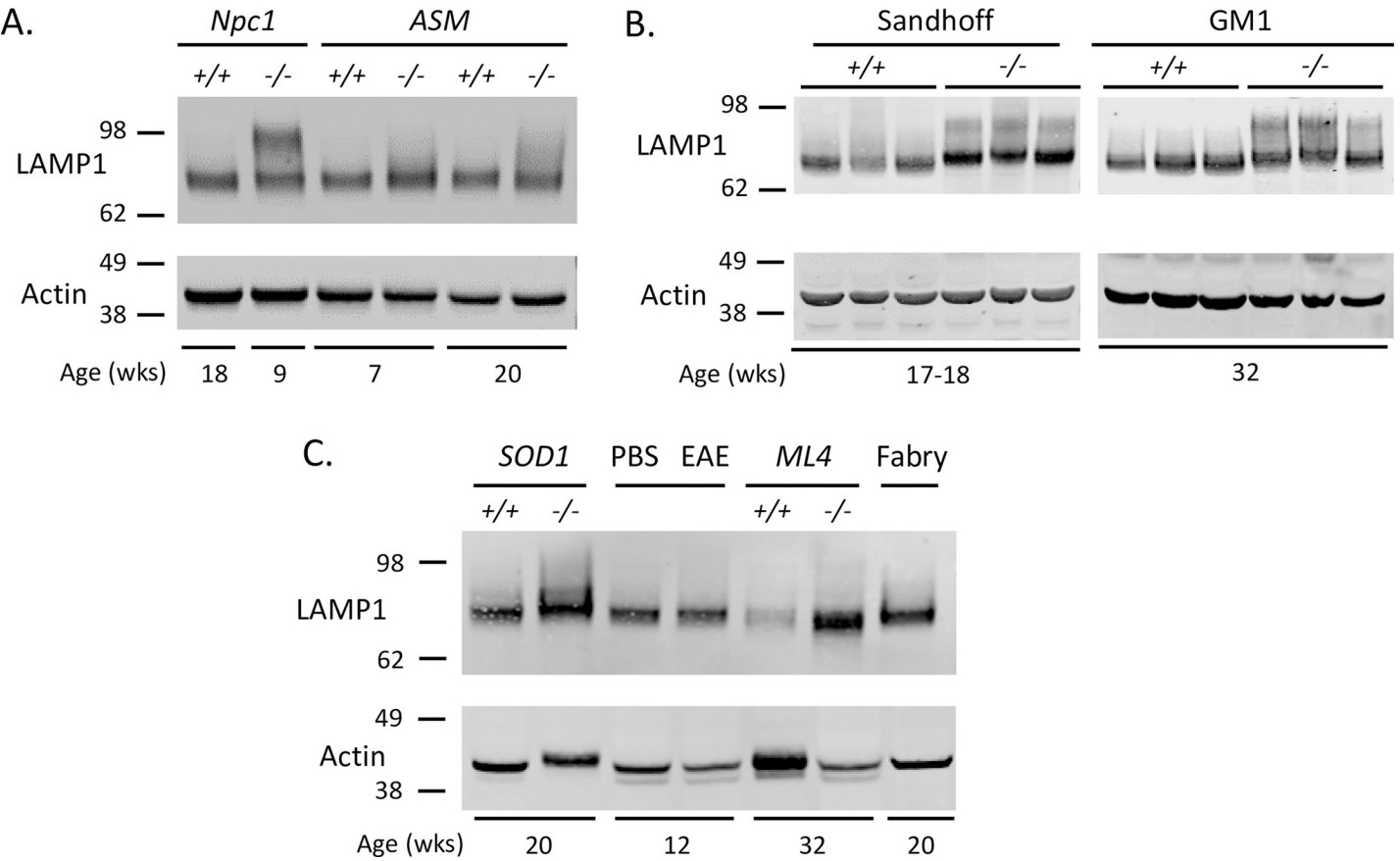

**Fig 7. Hyperglycosylated LAMP1 is unique to NPC diseases. A.** LAMP1 Western blot of whole cerebellar lysate from 18-week *Npc1*^+/+^, 9-week *Npc1*^−/−^ and 7- and 20-week-old acid sphingomyelinase KO mice (Niemann Pick Type A/B). **B.** LAMP1 Western blot of whole cerebellar lysate from control (+/+) and affected (-/-) ~18-week-old *Hexb* (Sandhoff) and 32-week-old *Glb1* (GM1) mice. **C.** LAMP1 Western blot of tissue lysate from control and mutant 20-week-old SOD1 mice (spinal cord), 12-week-old EAE mice and PBS control (cerebellum) and 32-week-old ML4 mice (hippocampus). Mouse cerebellar lysate from a 16-week-old Fabry mouse was used as a neuro-inflammation negative control.

2-hydroxypropyl-β-cyclodextrin (HPβCD), a drug currently in a phase 2b/3 clinical trial for the treatment of NPC1, was recently shown to delay the neuro-inflammation phenotype in *Npc1*^−/−^ mice [30]. Since hyperglycosylated LAMP1 is observed with the activated microglia during the neuro-inflammatory process in NPC, treatment of *Npc1*^−/−^ mice with HPβCD might be expected to reduce or eliminate the presence of hyperglycosylated LAMP1. Indeed, Western blot analysis showed that *Npc1*^−/−^ mice treated with HPβCD had only trace amounts of the hyperglycosylated LAMP1 compared to the *Npc1*^−/−^ mice treated with PBS (Fig 8A), consistent with the reduced neuro-inflammatory phenotype in the HPβCD treated mice [30]. Treatment of 7-week-old affected NPC1 mice with HPβCD for 2 weeks did not appear to significantly alter the disease progression (Fig 8B).

## LAMP1 in cerebellar tissues and CSF of individuals with NPC1

To investigate if the observations in mice could be seen in humans; tissue and CSF from NPC1 and control patients were analyzed. Western blot analysis of cerebellar tissue from NPC1 patients and age-matched controls showed that the predominant form of LAMP1 was smaller (~75–80 kDa) than the expected ~90–120 kDa size for LAMP1. This result is similar to the results obtained from the mouse model and indicates a possible conserved functional role for

LAMP1 in the CNS that does not require the extensive and more complex N-linked glycosylation that is found normally in the peripheral tissues. In NPC1 tissue, the presence of higher diffuse bands of LAMP1, indicative of hyperglycosylation, were observed in the cerebellum (Fig 9B); however, while LAMP1 was detected in the CSF of control and NPC1 samples, the predominant size in all cases was ~75–80 kDa, a size consistent with the hypo-glycosylated LAMP1 found in mouse and human CNS tissue. There was no difference in total levels of LAMP1 in the CSF between the two groups and a hyperglycosylated form of LAMP1 was not apparent in the CSF samples (S7 Fig).

## Discussion

In Niemann-Pick disease, type C (NPC), excess cholesterol accumulation in the endo-lysosomal compartment is caused by defects in one of two lysosomal proteins, either NPC1 or NPC2. These defects have far reaching consequences that affect cell function and viability. Understanding downstream pathways or other molecular components, that contribute to NPC pathology and might be amenable to intervention, is essential for developing therapies for NPC.

The role of glycosylation is one of many aspects of NPC that have been studied previously. The micro-heterogeneity in the size of NPC2 in late endosomes from $Npc1^{-/-}$ mice is due to differential N-linked glycosylation and appears to enhance the amount of NPC2 associated with an insoluble fraction after extraction [31]. The secretion of this form of NPC2 into bile may contribute to cholesterol transport in normal and disease states. Upon secretion it co-purifies with the pro-nucleating fraction of bile that can initiate cholesterol crystallization [32]. In the brain, the complexity of the glycosylation pattern of ApoE, a key player in cholesterol metabolism, is changed in the $Npc1^{-/-}$ mouse. The change in complex glycosylation and sialyation in ApoE correlates with Aβ (42) accumulation and this change may play a role in Aβ (42) clearance [33].

Recently it has been shown that the reduction of the glycocalyx in NPC1 fibroblasts in culture, by inhibiting O-linked glycosylation with benzyl-2-acetamido-2-deoxy-α-D-galactopyranoside (BADG), was shown to increase cholesterol efflux from these cells and reduce cholesterol storage in the lysosomes *in vitro* [8]. This suggests that the glycosylation state of LAMP1 (and LAMP2) may be an important factor in the overall mechanism of cholesterol efflux from the lysosome. A recent study found predominantly complex sialylated N-glycans in purified lysosomes from NPC1-null fibroblasts, in comparison to lysosomes from wild type cells, which had high-mannose N-glycans [9]. Such complexity gives rise to differences in the physico-chemical properties on the inner leaflet of the lysosome e.g. an increased net negative charge due to the sialic acid residues. This may add to the barrier effect of the glycocalyx and increase cholesterol retention in the lysosome. Pfeffer and colleagues [34] showed that LAMP1 and 2 can specifically bind cholesterol, and proposed that such binding may contribute to the handoff of cholesterol to NPC1 under normal conditions to contribute to cholesterol efflux from the lysosome [34]. In addition, a recent proteomics report on NPC patient cells and over-expression studies in HeLa cells have indicated a direct role of LAMP1 in alleviating cholesterol storage in these cells [35]. The concept of LAMPs functioning as a "store" of cholesterol is not without merit considering the stoichiometry and binding parameters of cholesterol to the LAMPs [34]. It is noteworthy, however, that the forms of LAMP1 used to bind cholesterol in the experiments by Pfeffer and colleagues were expressed in HEK293 or HeLa cells and as such would represent the "wild type" glycomic pattern of LAMP1. We speculate that the binding parameters of cholesterol to NPC1-cell derived LAMP1 might be different and could provide further insight into this mechanism in the disease state.

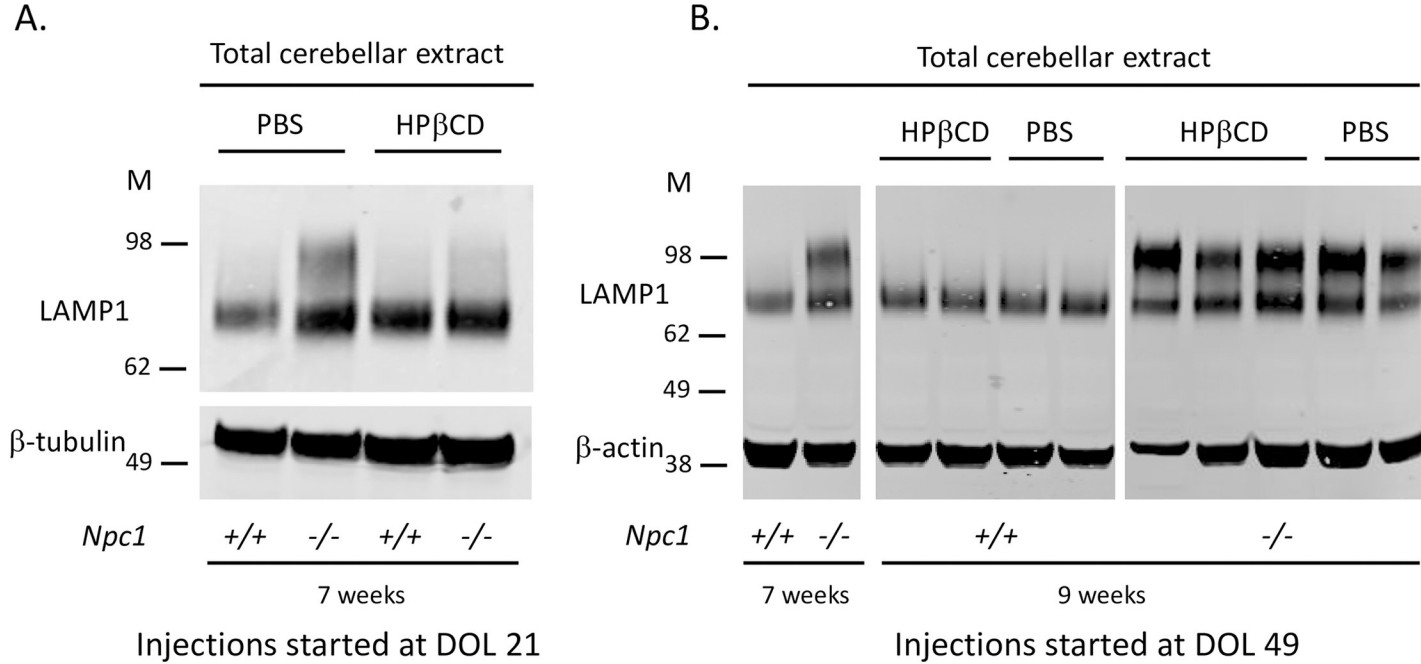

**Fig 8. Hyperglycosylated LAMP1 is regulated by 2-hydroxypropyl-β-cyclodextrin treatment. A.** LAMP1 and β-tubulin Western blot of total cerebellar extract from 7-week-old $Npc1^{+/+}$ and $Npc1^{-/-}$ mice treated with either 2-hydroxypropyl-β-cyclodextrin (HPβCD) or the vehicle control (PBS) every 3 days starting at 3 weeks of age. **B.** LAMP1 and β-tubulin Western blot of total cerebellar extract from 9-week-old $Npc1^{+/+}$ and $Npc1^{-/-}$ mice treated with either 2-hydroxypropyl-β-cyclodextrin (HPβCD) or the vehicle control (PBS) every 3 days starting at 7 weeks of age. Two untreated littermate controls ($Npc1^{+/+}$ and $Npc1^{-/-}$) were analyzed at 7 weeks of age. DOL, day of life.

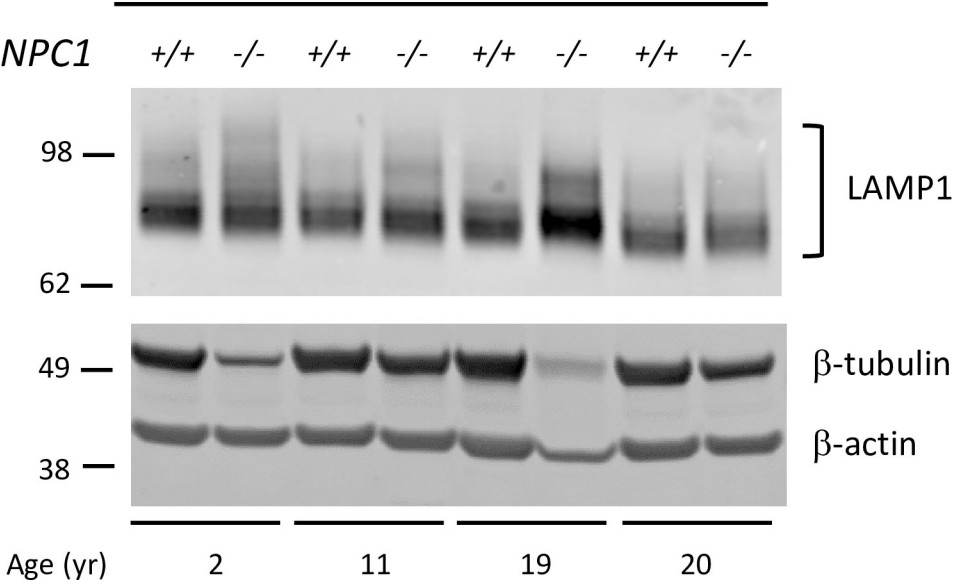

**Fig 9. Hyperglycosylated LAMP1 is present in NPC1 postmortem CNS tissue.** Western blot of human cerebellar tissue extracts from NPC1 (-/-) and age-matched controls (+/+). The Western blot shows LAMP1 with β-tubulin and β-actin as loading controls.

In characterizing the glycosylation state of LAMP1 in NPC1 mouse tissue, we observed differentially glycosylated forms across the lobules in the cerebellum of NPC1 mice that correlated spatiotemporally with disease progression and Purkinje neuron death (Fig 2) [1]. While the predominant hyperglycosylated form was subsequently identified as being present in activated microglia (discussed below), we observed a hypo-glycosylated, Endo H sensitive, form that was found in control ($Npc1^{+/+}$) and pre- and post-symptomatic $Npc1^{-/-}$ CNS tissues. It was noticeably smaller (~75 kDa) than that found in peripheral tissues (~90–120 kDa) (Fig 1); a finding that may represent a functional difference between peripheral and CNS LAMP1. The size of LAMP2 in CNS tissue is also smaller than seen in peripheral tissue ([18] and Fig 1), suggesting that a general difference might exist in the glycocalyx of lysosomes in the tissues of the CNS compared to peripheral tissues and would be predicted to result in a less complex glycocalyx specific to the CNS. LAMP2 and another lysosomal membrane protein, LIMP2, also showed an apparent general increase in the levels of these proteins in the diseased state (Fig 1), most likely because of the disease-related lysosomal/endosomal dysfunction and subsequent up-regulation to compensate.

The more prominent hyperglycosylated LAMP1 form was a surprising observation. Normally, LAMP1 in the cerebellum is ~75 kDa which we have shown is colocalized with Calbindin-positive cells (Purkinje neurons) and S100β-positive cells (Bergmann glia, BG) (Fig 4). This suggests that these cells under normal conditions may be more active in their (endo)lysosomal pathways than the other cells in the cerebellum. The exact role that BG play in NPC disease is unknown since the Purkinje neuron death appears cell-autonomous [36, 37]. However, poor morphological differentiation and function of BG in NPC [38], indicate they may contribute to the disease such as in glutamate buffering. The role LAMP1 plays in this function remains to be studied. The hyperglycosylated LAMP1 we observe is resistant to Endo H deglycosylation, demonstrating the presence of complex N-linked glycans; a pattern and size similar to LAMP1 found in peripheral tissues (Fig 1). However, infiltration of peripheral immune cells in the cerebellum of these mice has not been observed [30], suggesting that this peripheral glycosylation pattern is present in brain resident cells. LAMP2 and LIMP2 did not appear to have hyperglycosylated forms demonstrating that the differential glycosylation appeared to be specific for LAMP1 only.

Our immunofluorescence (IF) analysis showed LAMP1 was predominantly colocalized with activated microglia in $Npc1^{-/-}$ cerebella (Fig 5A), although there was some staining in GFAP positive reactive astrocytes (Fig 5C). This observation was validated by immunoprecipitation (IP) and FACS analysis of microglia from $Npc1^{-/-}$ cerebella and suggests that the hyperglycosylated LAMP1 that appears in the $Npc1^{-/-}$ CNS around the same time as Purkinje neuron degeneration, is present on the surface of activated microglia during neuro-inflammation. Microglia are brain resident immune cells that originate from the yolk sac during development [39]. It is therefore not surprising to find hyperglycosylated LAMP1 present in these cells (Fig 6 and S5 Fig), as they originate from a peripheral tissue. In addition, recent work by our group observed that $Lamp1$, but not $Lamp2$, is significantly upregulated ($Log_2Fold$, 1.08; Padj 4.66E-15), in activated microglia from $Npc1^{-/-}$ mice [30]. This is in contrast to microarray data showing reduced expression of $Lamp1$ in the whole cerebellum of $Npc1^{-/-}$ mice [40, 41]. These results suggest that LAMP1 may play a unique role in the neuro-inflammatory process of NPC. The lack of hyperglycosylated LAMP1 in affected CNS tissues from other mouse models of neurodegenerative diseases that exhibit neuro-inflammation renders our observation specific to NPC, suggesting a unique function associated with this protein in the context of NPC (Fig 7). Treatment of $Npc1^{-/-}$ mice with HPβCD every 3 days starting at day of life 21 prevented the appearance of hyperglycosylated LAMP1, while treatment every 3 days for 2 weeks starting at day of life 49, after disease onset, did not appear to have a significant effect. These results

suggest that it is important to start treatment early prior to disease phenotype onset for the treatment to have the best outcome. In human tissues, we did observe an increased level of hyperglycosylated LAMP1 in NPC1 cerebellar samples compared to age-matched control tissue. This is consistent with the activation of microglia observed in human NPC1 brain tissue [42]. We explored the potential of using hyperglycosylated LAMP1 as a CSF biomarker; however, we did not observe any significant differences between control and NPC1 CSF (S7 Fig).

In summary, our work has provided insight into the role that complex glycosylation on LAMP1 may play in NPC disease progression. Our finding highlights two possible modes by which this may occur; 1) through the interaction and signaling of LAMP1 on the surface of activated microglia and 2) through the change in the complex nature of glycosylation of LAMP1 in CNS cells. Further studies designed to address these questions are underway.

## Supporting information

**S1 Fig. Size estimation of mouse CNS LAMP1. (A)** Western blot of total cerebellar extract of $Npc1^{+/+}$ and $Npc1^{-/-}$ mice for LAMP1 (green) and heat shock protein 70 (HSP70) (upper red). Actin was used as a loading control.
(TIF)

**S2 Fig. Hyperglycosylated LAMP1 is composed of complex N-glycans. (A)** LAMP1 Western blot of 9-week $Npc1^{-/-}$ olfactory bulb (OB) and hippocampus (Hippo) lysate, treated with (+) and without (-) Endo H. **(B)** LAMP1 Western blot of 9-week $Npc1^{+/+}$ pancreas (Panc) and spleen lysate, treated with (+) and without (-) Endo H.
(TIF)

**S3 Fig. Immunofluorescence microscopy of midline sagittal cerebellar section of 9-week $Npc1^{+/+}$ mice.** Low magnification image of the whole cerebellar section stained with LAMP1 (red). Arrowheads indicate positive staining of the Purkinje cell layer. Granule cell layer (GCL); Molecular layer (ML).
(TIF)

**S4 Fig. Immunofluorescence microscopy of midline sagittal cerebellar section of 9-week $Npc1^{+/+}$ and $Npc1^{-/-}$ mice.** Higher magnification image of a lobule of $Npc1^{+/+}$ **(A)** and $Npc1^{-/-}$ **(B)** stained with LAMP1 (red). **A.** Arrows indicate positive LAMP1 staining of the Purkinje cell layer (closed arrows: Purkinje neuron soma; open arrows: other LAMP1 positive cells). **B.** LAMP1 staining is more disordered in the Purkinje cell layer of $Npc1^{-/-}$ cerebellum and has strong staining in the molecular layer. Granule cell layer (GCL); Molecular layer (ML).
(TIF)

**S5 Fig. Hyperglycosylated $Npc2^{-/-}$ LAMP1 is also present predominantly in microglia. (A & B)** LAMP1 Western blot of cell lysates from 16-week-old $Npc2^{+/-}$ and $Npc2^{-/-}$ mouse cerebella that underwent tissue dissociation and subsequent CD11b (microglia) or ASCA-2 (astrocyte) immunoprecipitation. **(A)** Lanes show the un-bound (UB) and bound (B) fractions of CD11b immunoprecipitation. **(B)** Lanes show the un-bound (UB) and bound (B) fractions of ASCA2 immunoprecipitation. Glutamine synthase (GS) and Glial fibrillary acid protein (GFAP).
(TIF)

**S6 Fig. Microglial LAMP1 is hyperglycosylated.** LAMP1 Western blot of microglia cell lysate treated with LPS or PBS (vehicle control).
(TIF)

**S7 Fig. Hyperglycosylated LAMP1 is not detected in NPC1 patient CSF. A.** Western blot of LAMP1 in CSF of NPC1 patient (lower panel) and adult healthy controls (upper panel). Coomassie Blue stain (indicated) was used to normalize LAMP1 to total protein. The LAMP1 band intensity was quantified and standardized to the total protein signal in the whole respective lane. **B.** The LAMP1 total protein ratios (mean +/- SD) for NPC1 patients and healthy controls is graphed to the right of the Western blots.
(TIF)

## Acknowledgments

We would like to thank Dr. Cristin Davidson, Albert Einstein College of Medicine, NY, for generating the ASM mouse tissue. We would also like to thank the NHLBI flow cytometry and the NICHD microscopy core facilities for assistance with the microglia and astrocyte isolation, and confocal microscopy, respectively. We would also like to express our appreciation to our clinical team who contributed to both the Natural History and phase 1/2a study and helped provide patient samples. Finally, we would like to express our appreciation to the guardians and patients who have participated in our clinical trials.

## Author Contributions

**Conceptualization:** Niamh X. Cawley, Christopher A. Wassif, Forbes D. Porter.

**Data curation:** Niamh X. Cawley, Caitlin Sojka, Anna T. Lyons, Elena-Raluca Nicoli, Forbes D. Porter.

**Formal analysis:** Niamh X. Cawley, Caitlin Sojka, Elena-Raluca Nicoli, Christopher A. Wassif, Forbes D. Porter.

**Investigation:** Niamh X. Cawley, Caitlin Sojka, Antony Cougnoux.

**Methodology:** Niamh X. Cawley, Caitlin Sojka, Antony Cougnoux.

**Project administration:** Niamh X. Cawley, Christopher A. Wassif.

**Resources:** Antony Cougnoux, Christopher A. Wassif, Forbes D. Porter.

**Supervision:** Niamh X. Cawley, Forbes D. Porter.

**Validation:** Niamh X. Cawley, Caitlin Sojka.

**Visualization:** Niamh X. Cawley, Caitlin Sojka, Antony Cougnoux.

**Writing – original draft:** Niamh X. Cawley, Caitlin Sojka.

**Writing – review & editing:** Niamh X. Cawley, Caitlin Sojka, Antony Cougnoux, Anna T. Lyons, Elena-Raluca Nicoli, Christopher A. Wassif, Forbes D. Porter.

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
