## [Decision Letter · Decision Letter 0]

20 Sep 2019

PONE-D-19-23845

Abnormal LAMP1 glycosylation may play a role in Niemann-Pick disease, type C pathology

PLOS ONE

Dear Dr. Porter,

Thank you for submitting your manuscript to PLOS ONE. After careful consideration, we feel that it has merit but does not fully meet PLOS ONE’s publication criteria as it currently stands. Therefore, we invite you to submit a revised version of the manuscript that addresses the points raised during the review process.

In particular, some loading and purity controls are missing and must be included.

The images in Fig 4 and 5 are poor quality, must be improved.

Please include the information requested by reviewer 2 in the materials and methods section.

Some conclusions are not fully supported by data; therefore, they should be re-elaborated or further data should be provided.

The discussion can be improved by re-phrasing some sentences and discussing data already published, showing altered glycosylated proteins in NPC.

We would appreciate receiving your revised manuscript by Nov 04 2019 11:59PM. To enhance the reproducibility of your results, we recommend that if applicable you deposit your laboratory protocols in protocols.io, where a protocol can be assigned its own identifier (DOI) such that it can be cited independently in the future. For instructions see: http://journals.plos.org/plosone/s/submission-guidelines#loc-laboratory-protocols

We look forward to receiving your revised manuscript.

Kind regards,

Andrea Dardis, Ph.D.

Academic Editor

PLOS ONE

Journal Requirements:

Reviewers' comments:

Reviewer's Responses to Questions

**Comments to the Author**

1. Is the manuscript technically sound, and do the data support the conclusions?

Reviewer #1: Partly

Reviewer #2: Partly

2. Has the statistical analysis been performed appropriately and rigorously? 

Reviewer #1: Yes

Reviewer #2: Yes

3. Have the authors made all data underlying the findings in their manuscript fully available?

Reviewer #1: Yes

Reviewer #2: Yes

4. Is the manuscript presented in an intelligible fashion and written in standard English?

Reviewer #1: Yes

Reviewer #2: Yes

5. Review Comments to the Author

Reviewer #1: In this manuscript Cawley et al. show an altered glycosylation pattern of LAMP-1 which correlates with disease pathology. In many figures loading controls are missing and should be included (details below). The findings are interesting and support previous studies showing altered glycosylated patterns in NPC. Some relevant papers have been discussed; however other key papers must be included in the discussion. For instance, it has been shown that NPC2 is overexpressed in Npc1-/- cells and tissues and present changes in glycosylation patterns (PMID: 15896196, PMID: 16374838). Furthermore, ApoE also shows altered glycosylation patter in NPC (PMID: 20070866).

To facilitate the understanding of the article, some changes are suggested:

Figure1:

Figure 1A doesn’t add much information, it can be supplementary.

Figure 1B: Loading control is missing.

An interesting question that the authors can address is how is LAMP-1 overexpressed? Is it transcriptionally regulated? This can be easily analyzed by q-PCR analysis and would complement the Figure 1B.

Figure 2:

If actin is added to Figure 2B, then 2A doesn’t add any info. It is highly recommended to add loading controls.

Figures 4 and 5: Both Figures should be merged. High magnification images are necessary. The images shown in supplementary Fig 2S are more informative than the currently shown in fig 4. Please replace them and show comparative images for Npc1-/- tissues.

Figure 6: Nice idea; however, some controls are missing. A control showing the purity of the isolations is necessary, similarly to what is shown in Fig 3S. In addition to GFAP a microglial marker should be analyzed (CD11b or others)

Figure 7: nice work here, Actin is missing in Fig 7C.

Reviewer #2: This study identifies and characterizes the presence of a high level of glycosylated form of LAMP1 in the cerebellum and other brain regions of Niemann Pick C1 mice. Hyperglycosylated LAMP1 appears to be associated to disease progression and is rescued by the treatment with HPβCD started at pre-symptomatic stages of the disease. Interestingly, a glycosylated form of LAMP1 is also observed in cerebellar samples obtained from NPC patients, but not in those obtained from patients affected by different neurodegenerative diseases.

In addition, the study provides evidence that this post-translationally modified form of LAMP1 is linked to microglia-mediated neuroinflammation.

Experiments presented are well conceived and data provided add a novel and relevant contribution to the knowledge of pathogenic mechanisms of the NPC disease. There are, however, some criticisms that the authors should address to strengthen the major findings of their study.

Specific issues.

Figure 1. Given that the estimated MW of the brain form of LAMP1 is ≈ 75 kDa, the lack of a MW reference standard near this range size appears a bit peculiar. An additional reference standard between 62 and 98 kDa would strengthen this estimate.

The authors discuss the findings displayed by panel B in terms of quantitative variation of protein abundance. However, this inference is not circumstantiated with loading controls.

Figure 2. Western blot displayed in panel B and quantitative analyses of panel C refer to protein extracted from anterior, middle and posterior part of the cerebellum. However, no detail is given on how these compartments were separated one from each other (i.e. which lobules were included in each region?). The procedure should be described under materials and methods.

Figure 3 (A, B). The treatment with N-Glycanase and EndoH appears to be accompanied by a sensible decrease of protein abundance, as suggested by the weaker intensity of bands corresponding to treated extract loadings. The authors should comment on this. It might depend on antibody affinity for modified/unmodified protein forms, respectively. Once again, loading controls are missing, whereas their presence would contribute to clarify this issue.

Figure 4. LAMP1 immunostaining of panel B is consistent with the presence of LAMP1 in Bergmann glia (BG), as stated by the authors. However the merged picture shown in panel C, does not properly represent this feature, showing a very faint and sparse red staining (LAMP1) that is masked by the strong Calbindin staining of Purkinje cells (PC), perhaps. This is somehow inconsistent with the picture displayed by panel A of Fig. 5, which clearly shows the presence of LAMP1 in PC somata. To my opinion, information of Figure 4 is somehow redundant in light of data displayed in Fig. 5.

Figure 5. Cerebellar lobules displayed in the various panels are not comparable, whereas the typical spatiotemporal spread out of the NPC disease within the cerebellum makes this issue worth of careful attention. I strongly encourage the authors to consistently select the same/similar (anterior, middle, posterior) lobules for displaying.

The statement: “There was weak co-staining with GFAP, a marker of reactive astrocytes”, is only partly correct, since GFAP also stains Bergmann glia (BG). Noteworthy, besides astrocytosis, previous studies have reported a number of anomalies of BG in mouse models of NPC since early stages of the disease. In light of the presence of LAMP1 in BG of wild-type mice, the sparse/decreased LAMP1 presence in BG of NPC mice could be a sign of the disease. This possibility should be taken into account by the authors.

Figure 6. The emphasis that the authors deserve to the intermediate sized LAMP1, to my opinion, is not supported by the poor definition of this band.

Minor points

The discussion is a bit too long and the overall readability of the manuscript can be improved by re-phrasing some sentences. As an example, see: “The disease is autosomal recessive, where patients………….to other cellular organelles”.

6. PLOS authors have the option to publish the peer review history of their article (what does this mean?). If published, this will include your full peer review and any attached files.

Reviewer #1: Yes: Andres D Klein

Reviewer #2: No

---

## [Author Response · Author response to Decision Letter 0]

16 Dec 2019

Reviewer #1: In this manuscript Cawley et al. show an altered glycosylation pattern of LAMP-1 which correlates with disease pathology. In many figures loading controls are missing and should be included (details below). The findings are interesting and support previous studies showing altered glycosylated patterns in NPC. Some relevant papers have been discussed; however other key papers must be included in the discussion. For instance, it has been shown that NPC2 is overexpressed in Npc1-/- cells and tissues and present changes in glycosylation patterns (PMID: 15896196, PMID: 16374838). Furthermore, ApoE also shows altered glycosylation patter in NPC (PMID: 20070866).

Thank you for reviewing our paper. Our apologies for omitting several papers. We have now included them and integrated their information in the Discussion section (lines 428-436).

To facilitate the understanding of the article, some changes are suggested:

Figure1:

Figure 1A doesn’t add much information, it can be supplementary. We are not aware of any previously reported specific size determination of LAMP1 in different brain regions of the mouse. When we first performed the Western blot, we were surprised to see it so “small” compared to the size reported in most of the literature, which usually describes the peripheral form of LAMP1. A smaller LAMP1 sized band was seen in the original LAMP1 KO paper, but it’s size was not discussed and was only used as a control. We would therefore like to keep this part of Fig 1 as is, since it represents a clear size distinction between peripheral and CNS LAMP1 that may be related to function.

Figure 1B: Loading control is missing. We have added the blot probed for actin as a loading control. This blot has now been added to the Fig 1B.

An interesting question that the authors can address is how is LAMP-1 overexpressed? Is it transcriptionally regulated? This can be easily analyzed by q-PCR analysis and would complement the Figure 1B. This is a good question but has been addressed previously by us and others as part of microarray and RNA-seq datasets. Specifically, Liao et al, (PMC2848886) showed a slight reduction (Log2(FC) = -0.071) of LAMP1 in 3 week old Npc1-/- cerebellum versus Npc1+/+. In addition, Vasquez et al, (PMC3245218) found it to be significantly reduced (log2(FC) = -2.12) in the cerebellum of 7-8 weeks old Npc1-/- mice compared to Npc1+/+. Our previous RNA-seq data (Cougnoux et al, PMC5985727) showed that LAMP1 mRNA was increased ~2-fold in microglia from the cerebellum of Npc1-/- mice compared to Npc1+/+ (log2(FC) = 1.08). These data suggest an overall reduction in LAMP1 mRNA expression in Npc1-/- in the cerebellar tissue but with a specific increase in the microglia. We have now included this information in the Discussion (lines 496-499).

Figure 2:

If actin is added to Figure 2B, then 2A doesn’t add any info. It is highly recommended to add loading controls. Unfortunately, we do not have these samples for 2B anymore. The blot in Fig 2B was meant as a representative blot to visualize the bands. However, the complete data reported in the bar graph is a measure of the ratio of the two bands within each lane and hence are normalized within the lane. In the absence of these samples to re-run, we hope our justification is acceptable to have the figure stand as is.

Figures 4 and 5: Both Figures should be merged. High magnification images are necessary. The images shown in supplementary Fig 2S are more informative than the currently shown in fig 4. Please replace them and show comparative images for Npc1-/- tissues. As requested, we have now repeated the IHC and analyzed it by confocal microscopy and have included comparative images for Npc1-/-. The high magnification confocal images of LAMP1 with Calbindin (Purkinje cells) or S100��(Bergmann glia) in control and Npc1-/- mice (lobule I) is now new figure 4. We have placed the old figure 4 (low magnification of the whole cerebellum staining of LAMP1) as the new supplementary figure 3. In addition, we placed the higher magnification image of LAMP1 staining only (part of old figure 4) as new supplemental figure 4. 

We have also modified Figure 5 to include only co-staining of Iba, Olig2 and GFAP with LAMP1 (since calbindin and S100� is now presented in new figure 4). We felt that merging both figure 4 and 5 would make the figure too complex and we would like to keep it as 2 separate figures. We hope that this layout is acceptable to the reviewer.

Figure 6: Nice idea; however, some controls are missing. A control showing the purity of the isolations is necessary, similarly to what is shown in Fig 3S. In addition to GFAP a microglial marker should be analyzed (CD11b or others). We have now repeated the microglia purification and analyzed the bound and unbound fractions from both Npc1+/+ and Npc1-/- cerebella. We probed for CD11b and confirmed that all the CD11b staining was in the bound fraction. Analysis for an astrocyte marker, glutamine synthase (GS), showed the opposite i.e. all GS was recovered in the unbound fraction and none in the bound fraction. The hyper-glycosylated LAMP1 was recovered almost exclusively in the bound fraction consistent with its presence in microglia.

We have modified the figure to focus on the microglia purification only and have modified the text. We did not repeat the astrocyte purification because the GLAST antigen used to pull down the cells is susceptible to papain treatment used for the cell dissociation and it would not work with the current Miltenyi IP kit. However, the supplementary figure 5 showing purification of astrocytes from Npc2 mouse cerebella demonstrates that the hyperglycosylated LAMP1 is not present in these cells.

Figure 7: nice work here, Actin is missing in Fig 7C. We have re-run these samples and included �-actin as a loading control. This blot has now been added to the Fig 7C.

 

Reviewer #2: This study identifies and characterizes the presence of a high level of glycosylated form of LAMP1 in the cerebellum and other brain regions of Niemann Pick C1 mice. Hyperglycosylated LAMP1 appears to be associated to disease progression and is rescued by the treatment with HPβCD started at pre-symptomatic stages of the disease. Interestingly, a glycosylated form of LAMP1 is also observed in cerebellar samples obtained from NPC patients, but not in those obtained from patients affected by different neurodegenerative diseases.

In addition, the study provides evidence that this post-translationally modified form of LAMP1 is linked to microglia-mediated neuroinflammation.

Experiments presented are well conceived and data provided add a novel and relevant contribution to the knowledge of pathogenic mechanisms of the NPC disease. There are, however, some criticisms that the authors should address to strengthen the major findings of their study.

We thank you for taking the time to assess our work. We have addressed your comments and made the changes where necessary. Please see below.

Specific issues.

Figure 1. Given that the estimated MW of the brain form of LAMP1 is ≈ 75 kDa, the lack of a MW reference standard near this range size appears a bit peculiar. An additional reference standard between 62 and 98 kDa would strengthen this estimate. Thank you, yes, this is more accurate. We have now re-run several samples and probed the blot for Heat Shock Protein 70 (HSP70). This 70kDa HSP70 band is clearly above the 62kDa marker but still slightly below the brain form of LAMP1. While the accuracy of the mini-gels in this region of the gel has its limits we believe that assessing the brain form of LAMP1 as approximately 75 kDa is reasonable. We have included this blot as part of the Supplementary information (S1 Fig) and have referred to it in the main text (line 242-243).

The authors discuss the findings displayed by panel B in terms of quantitative variation of protein abundance. However, this inference is not circumstantiated with loading controls. We have re-run these samples and probed for �-tubulin as a loading control. The blot shows similar protein loaded per lane and has now been added to the Fig 1B.

Figure 2. Western blot displayed in panel B and quantitative analyses of panel C refer to protein extracted from anterior, middle and posterior part of the cerebellum. However, no detail is given on how these compartments were separated one from each other (i.e. which lobules were included in each region?). The procedure should be described under materials and methods. Our apologies for the oversight. Lobules I to V were dissected as anterior, IV to VIII as middle and IX and X as posterior. The procedure has now been added to the Methods section of the manuscript (line 124-128).

Figure 3 (A, B). The treatment with N-Glycanase and EndoH appears to be accompanied by a sensible decrease of protein abundance, as suggested by the weaker intensity of bands corresponding to treated extract loadings. The authors should comment on this. It might depend on antibody affinity for modified/unmodified protein forms, respectively. Once again, loading controls are missing, whereas their presence would contribute to clarify this issue. We agree with the reviewer. In addition to antibody affinity as suggested, we cannot rule out the low levels of protease activity present that may have contributed to the lower recovery. In addition, the loss due to sticking to the tube (of the deglycosylated form) after the long incubation time may contribute. However, our focus for this experiment was qualitative rather than quantitative. To that end, the qualitative shift in size is strong evidence of its unique N-linked glycosylation nature. As requested, we have commented on this in the manuscript as requested (line 291-292).

Figure 4. LAMP1 immunostaining of panel B is consistent with the presence of LAMP1 in Bergmann glia (BG), as stated by the authors. However the merged picture shown in panel C, does not properly represent this feature, showing a very faint and sparse red staining (LAMP1) that is masked by the strong Calbindin staining of Purkinje cells (PC), perhaps. This is somehow inconsistent with the picture displayed by panel A of Fig. 5, which clearly shows the presence of LAMP1 in PC somata. To my opinion, information of Figure 4 is somehow redundant in light of data displayed in Fig. 5. As requested by reviewer 1 and from your comments here, we have now repeated the IHC by confocal microscopy and have made a new figure 4 to include Calbindin/LAMP1 and S100b/LAMP1 in control and Npc1-/- mice. We have also rearranged some images to supplementary for clarity. The results are consistent with our previous data, but it also clarifies for us that other cells present are staining with LAMP1 and not calbindin or S100b. We have included this in the results (lines 309-315) section.

Figure 5. Cerebellar lobules displayed in the various panels are not comparable, whereas the typical spatiotemporal spread out of the NPC disease within the cerebellum makes this issue worth of careful attention. I strongly encourage the authors to consistently select the same/similar (anterior, middle, posterior) lobules for displaying. We have now identified and labelled the lobules shown in the panels of Fig 5. In general, our goal was to focus on the LAMP1 staining in Npc1-/- in the context of other known cells present. In this case the images shown primarily contained lobules 4/5 for oligodendrocyte, astrocytes, microglia. 

The statement: “There was weak co-staining with GFAP, a marker of reactive astrocytes”, is only partly correct, since GFAP also stains Bergmann glia (BG). Noteworthy, besides astrocytosis, previous studies have reported a number of anomalies of BG in mouse models of NPC since early stages of the disease. In light of the presence of LAMP1 in BG of wild-type mice, the sparse/decreased LAMP1 presence in BG of NPC mice could be a sign of the disease. This possibility should be taken into account by the authors. We agree with the reviewer’s suggestion and have changed the sentence to include “a marker of reactive astrocytes including Bergmann glia” (line 327-328). As requested by reviewer 1, we performed confocal microscopy for calbindin/LAMP1 and s100b/LAMP1 on both control and mutant mice. We do not know why LAMP1 is so concentrated in Purkinje cells (PCs) and Bergmann glia (and possibly other cells) in the PC layer of control mice; nor about its contribution to the disease in these cells. We cannot really say that LAMP1 staining (or lack) in BG may be a sign of the disease. Our focus in this paper with respect to LAMP1 was the microglia, however, in light of the poor morphological differentiation and function of BG (Caporali et al, 2016, Acta Neuropathol Commun), we have included a comment in the text about BG function (lines 477-481).

Figure 6. The emphasis that the authors deserve to the intermediate sized LAMP1, to my opinion, is not supported by the poor definition of this band. We agree with the reviewer and have removed all references to this observation. 

Minor points

The discussion is a bit too long and the overall readability of the manuscript can be improved by re-phrasing some sentences. As an example, see: “The disease is autosomal recessive, where patients………….to other cellular organelles”. We have now edited the manuscript and Discussion to be more readable. We have removed the second last paragraph discussing the possible idea about the poly-LacNAc moieties as this is all speculation. However, reviewer 1 requested we add something about other glycosylated proteins in NPC. In all we have reduced the length of the Discussion.

---

## [Decision Letter · Decision Letter 1]

31 Dec 2019

Abnormal LAMP1 glycosylation may play a role in Niemann-Pick disease, type C pathology

PONE-D-19-23845R1

Dear Dr. Porter,

We are pleased to inform you that your manuscript has been judged scientifically suitable for publication and will be formally accepted for publication once it complies with all outstanding technical requirements.

With kind regards,

Andrea Dardis, Ph.D.

Academic Editor

PLOS ONE

Additional Editor Comments (optional):

Reviewers' comments:

Reviewer's Responses to Questions

**Comments to the Author**

1. If the authors have adequately addressed your comments raised in a previous round of review and you feel that this manuscript is now acceptable for publication, you may indicate that here to bypass the “Comments to the Author” section, enter your conflict of interest statement in the “Confidential to Editor” section, and submit your "Accept" recommendation.

Reviewer #1: All comments have been addressed

Reviewer #2: All comments have been addressed

2. Is the manuscript technically sound, and do the data support the conclusions?

Reviewer #1: Yes

Reviewer #2: Yes

3. Has the statistical analysis been performed appropriately and rigorously? 

Reviewer #1: Yes

Reviewer #2: Yes

4. Have the authors made all data underlying the findings in their manuscript fully available?

Reviewer #1: Yes

Reviewer #2: Yes

5. Is the manuscript presented in an intelligible fashion and written in standard English?

Reviewer #1: Yes

Reviewer #2: Yes

6. Review Comments to the Author

Reviewer #1: The new figures have made the manuscript much better than the first one.

All the comments were property addressed.

Happy Holidays!

Reviewer #2: (No Response)

7. PLOS authors have the option to publish the peer review history of their article (what does this mean?). If published, this will include your full peer review and any attached files.

Reviewer #1: Yes: Andres D Klein

Reviewer #2: No

---

## [Editor Report · Acceptance letter]

14 Jan 2020

PONE-D-19-23845R1 

Abnormal LAMP1 glycosylation may play a role in Niemann-Pick disease, type C pathology 

Dear Dr. Porter:

I am pleased to inform you that your manuscript has been deemed suitable for publication in PLOS ONE. Congratulations! Your manuscript is now with our production department. 

With kind regards,

on behalf of

Dr. Andrea Dardis 

Academic Editor

PLOS ONE